# The Role of the Dorsolateral Prefrontal Cortex in the Production and Comprehension of Phonologically and Semantically Related Words

**DOI:** 10.3390/brainsci13071113

**Published:** 2023-07-22

**Authors:** Lindsay K. Butler, Meredith Pecukonis, De’Ja Rogers, David A. Boas, Helen Tager-Flusberg, Meryem A. Yücel

**Affiliations:** 1Speech, Language & Hearing Sciences, University of Connecticut, Storrs, CT 06269, USA; 2Psychological & Brain Sciences, Boston University, Boston, MA 02215, USA; mpecukon@bu.edu (M.P.); htagerf@bu.edu (H.T.-F.); 3Biomedical Engineering, Boston University, Boston, MA 02215, USA; dejar14@bu.edu (D.R.); dboas@bu.edu (D.A.B.);

**Keywords:** dorsolateral prefrontal cortex, lexical selection, fNIRS, semantics, phonology, blocked cyclic naming

## Abstract

Previous studies suggest that producing and comprehending semantically related words relies on inhibitory control over competitive lexical selection which results in the recruitment of the left inferior frontal gyrus (IFG). Few studies, however, have examined the involvement of other regions of the frontal cortex, such as the dorsolateral prefrontal cortex (DLPFC), despite its role in cognitive control related to lexical processing. The primary objective of this study was to elucidate the role of the DLPFC in the production and comprehension of semantically and phonologically related words in blocked cyclic naming and picture–word matching paradigms. Twenty-one adults participated in neuroimaging with functional near-infrared spectroscopy to measure changes in oxygenated and deoxygenated hemoglobin concentrations across the bilateral frontal cortex during blocked cyclic picture naming and blocked cyclic picture–word-matching tasks. After preprocessing, oxygenated and deoxygenated hemoglobin concentrations were obtained for each task (production, comprehension), condition (semantic, phonological) and region (DLPFC, IFG). The results of pairwise *t*-tests adjusted for multiple comparisons showed significant increases in oxygenated hemoglobin concentration over baseline in the bilateral DLPFC during picture naming for phonologically related words. For picture–word matching, we found significant increases in oxygenated hemoglobin concentration over baseline in the right DLPFC for semantically related words and in the right IFG for phonologically related words. We discuss the results in light of the inhibitory attentional control over competitive lexical access theory in contrast to alternative potential explanations for the findings.

## 1. Introduction

Producing and comprehending words involves encoding and decoding the meaning of the word, or the semantic representation; and the associated string of sounds, or the phonological representation. A well-established frontotemporal language network supports semantic and phonological processing during lexical selection [1,2], inter alia. Additional brain mechanisms complement and support the canonical language network, such as motor and sensory representations, nonverbal memory structures, social cognition and cognitive control, all of which play a fundamental role in natural language communication, which may otherwise break down without these fundamental supports [3,4]. The dorsolateral prefrontal cortex (DLPFC) is considered part of the multiple demand system, a domain-general frontoparietal network functioning to assert superordinate control over various cognitive tasks. Though the DLPFC is not typically considered part of the canonical language network, it is anatomically adjacent and functionally connected, and has been associated with a range of language functions, including lexical selection during word production and comprehension, the focus of the current study. The overarching goal of this paper is to elucidate the specific role of the DLPFC by comparing activation during the production and comprehension of semantically and phonologically related words.

### 1.1. Effects of Semantic Relatedness on Word Production

Several early studies showed that naming pictures of semantically related words (e.g., words of the same category, as in “apple”, “pear”, “banana”, “peach”), resulted in increased naming latencies compared to naming pictures of unrelated words [5,6,7,8,9]; but see [10,11]. Going beyond behavioral measures, several studies used fMRI to investigate neural activation in picture naming for semantically related compared to unrelated words. When people named pictures of semantically related words, the investigators found greater activation in the left IFG and left temporal regions, areas of the canonical language network, compared to naming unrelated words [12,13]. These studies attributed a central role to the left IFG in resolving semantic competition due to coactivation during lexical selection, though other approaches suggest that task-based incremental learning is at play; see, e.g., [14].

### 1.2. Semantic Relatedness during Word Comprehension

Most behavioral and neuroimaging studies on competitive lexical selection due to semantic relatedness focus on picture naming. Fewer studies have investigated whether these effects extend to lexical comprehension during picture–word matching. Several behavioral studies have reported increased picture–word matching latencies for words with semantically related foils compared to unrelated foils [12,15,16]. Based on the findings that semantic relatedness leads to increased naming latencies (presumably reflective of greater cognitive effort) for lexical processing in both production and comprehension, interference is assumed to originate at the semantic level in both picture naming and picture–word matching. Few studies, however, have directly compared production and comprehension tasks. Furthermore, no studies, to the best of our knowledge, have investigated the neural correlates of semantic relatedness during lexical comprehension. The rationale for directly comparing lexical production and comprehension is that this approach will provide important evidence to clarify whether the role of the DLPFC is specific to inhibiting semantically related competitor words during lexical selection, or if it plays an alternative and potentially more domain-general role. If the DLPFC was implicated in both lexical production and comprehension, this would provide evidence for the inhibitory control competitive lexical selection approach to the role of the DLPFC in language. If, however, the DLPFC was not involved in both domains, alternative roles of DLPFC in lexical processing should be explored.

### 1.3. Effects of Phonological Relatedness on Word Production

Compared to studies on semantic relatedness in lexical production, fewer studies have examined the effects of phonological relatedness on naming latencies or neural activation. Schnur and colleagues [13] investigated naming errors and canonical language network activation during the naming of phonologically related words compared to unrelated words. While they did not directly compare semantic and phonological relatedness (instead, comparing semantically related to unrelated and phonologically related to unrelated), naming phonologically related words did not result in significant activation in the left IFG compared to unrelated words (while semantically related words did and were associated with more naming errors). The authors also discussed the observation that neither semantically nor phonologically related words were associated with significant activation in left temporal regions of the canonical language network compared to unrelated words. While their focus was on the canonical language network, a whole-brain analysis identified frontal activation that extended to the left middle frontal gyrus, the locus of the DLPFC. The lack of left IFG activation during the naming of phonologically related pictures and the whole-brain analysis of Schnur et al. [13] suggests that a closer examination of the role of other regions of the frontal cortex in lexical processing, specifically DLPFC, is needed, and motivates the current study.

Several studies have used EEG to investigate semantic and phonological relatedness with a focus on the time course of semantic relative to phonological processing during lexical selection in picture naming [17,18]. In terms of reaction times, naming semantically related words led to longer naming latencies (relative to naming unrelated words), while naming phonologically related words led to shorter naming latencies (relative to naming unrelated words). Using ERP, semantically related word naming was associated with a positive component around 200 ms, while naming phonologically related words was associated with a positive component around 350 ms. These findings align with the locus of lexical selection based on meta-analyses of the spatial and temporal signatures of word production [19,20]. Because these studies reported positive ERP components for semantically and phonologically related words, they concluded that these effects are task-related and the result of top-down cognitive processes, rather than stemming from competitive lexical selection in semantic memory (assumed to be localized to the left IFG and left temporal regions). In relation to the current study, the positive ERP components for semantically and phonologically related stimuli suggest that a closer look at brain response to semantic vs. phonological relatedness is needed, as well as a closer look across the domains of lexical production and comprehension, as no previous studies have investigated phonological relatedness in word comprehension.

### 1.4. The Role of Executive Function

Executive functions are cognitive processes that are necessary for goal-oriented tasks such as language production and comprehension. They include the ability to hold and change attentional focus, temporarily maintain information in working memory, organize information, self-monitor, inhibit responses, think flexibly and plan future actions. These processes are crucial for novel or complex tasks involving attention, as they influence nearly every aspect of cognition. One of the most important structures supporting executive function is the prefrontal cortex (see [21] for an overview). Specifically, DLPFC is a brain area neuroanatomically situated in the middle frontal gyrus (Broadmann areas 46 and 9 [22,23]). The DLPFC is considered part of the multiple-demand system, a domain-general frontoparietal network in which it may function to assert superordinate cognitive control over various cognitive tasks. These tasks include executive control functions such as task switching and task-set reconfiguration, prevention of interference, inhibition, planning and working memory, e.g., [24,25,26]. While the DLPFC has not been traditionally considered part of the canonical language network, it is related by adjacency and connectivity, and has been shown to be activated for some speech and language functions [27,28] such as word naming in the context of high-name-agreement words versus low-name-agreement words [29].

In terms of the relationship between executive function and language, the DLPFC has frequently been implicated in a range of executive functions necessary for complex language tasks (see [28] for an overview). Broad functions, such as executive control and working memory, as well as more specific functions, such as detecting novelty of incoming information [30,31] and conflict detection, resolution and adaptation [32], have been the focus of studies exploring the connection between the DLPFC and language. Another promising potential link between the DLPFC and the canonical language network is that elaborated timing functions are required for articulatory motor activity to produce a smooth speech signal integrated with the time requirements of cognitive processes such as lexical access, particularly in the case of difficult lexical access [28]. A related clinical study showed that stutterers exhibit a reduced activation of DLPFC in conflict tasks, which may be reflective of reduced readiness to execute a sequence of timed motor responses [33].

The primary objective of the current study is to elucidate the role of the DLPFC in the production and comprehension of semantically and phonologically related words. If we find that the DLPFC is active while participants are producing and comprehending semantically related words, but not phonologically related words, this evidence would suggest that the DLPFC plays an important role in executive control by inhibiting competing words in semantic memory (and no role associated with phonological processes). If however, we find that the DLPFC is active only in word production (for semantically and phonologically related stimuli) but not word comprehension, then the inhibitory control over lexical access role of the DLPFC would not be supported, as lexical access is necessary for both word production and comprehension.

### 1.5. Rationale for fNIRS Imaging

In the current study, we use functional near-infrared spectroscopy (fNIRS) to estimate neural activation by measuring changes in oxygenated and deoxygenated hemoglobin concentrations in response to tasks compared to the baseline. fNIRS has several advantages in the context of neuroimaging during speech and language processes. First, it is more tolerant of motion than fMRI, making it better suited to tasks that involve speaking [34]. Second, fNIRS is more amenable to clinical applications, as repeated measurements in close time intervals are affordable and well tolerated [35,36]. Picture naming and picture–word matching tasks are powerful tools in the evaluation of language impairment across etiologies (e.g., dementia, aphasia, developmental language disorder), so a more complete characterization of DLPFC activation during these tasks will help improve the assessment and intervention of cognitive and communication disorders.

### 1.6. Current Study

To the best of our knowledge, no studies to date have directly compared the effects of semantic and phonological relatedness on word production and comprehension. This is the overarching goal of the current study. We use the blocked cyclic naming [7] and the analogous blocked cyclic picture–word-matching tasks, which have increasingly been used to investigate lexical selection mechanisms and the associated neural correlates [13,17,18,29].

## 2. Methods

### 2.1. Participants

Twenty-one adults (13 female) between the ages of 18 and 27 (M = 21.86, SD = 2.85) were recruited from the Boston University community. The study was conducted according to the guidelines of the Declaration of Helsinki and approved by the Institutional Review Board of Boston University (IRB #4502). Informed consent was obtained from all subjects involved in the study. Nineteen participants were right-handed, and two were left-handed. The study lasted no longer than 2 h. Participant race and ethnicity are reported in Table 1.

Twenty–three adults were recruited, but data from two participants were not included in the final analysis. One participant was excluded due to an unreported visual impairment that was later disclosed. The second participant was excluded due to poor optode contact with the scalp related to time restrictions on human-to-human contact during the COVID–19 return–to–research phase.

### 2.2. Task Design and Procedures

The experimental tasks were designed and administered using PsychoPy [37]. Two experimental tasks (blocked cyclic picture naming and blocked cyclic picture–word matching) were counterbalanced by condition (within the task) and by task (within the study as a whole) and arranged into lists in a Latin squares design, such that different participants started the study with different tasks and runs to avoid any potential confounding effects of task or order of tasks. The picture stimuli for both tasks were selected from the Bank of Standardized Stimuli [38], which are normed for a range of visual and linguistic features. The conditions were counterbalanced into four runs (two runs of each task), each lasting approximately four minutes. Participants were instructed to remain as still as possible during each four-minute run. For the picture naming task, participants were instructed to say one word to name the picture on the screen. For the picture–word matching task, participants were instructed to touch the picture on the screen that matched the word they heard. For both tasks, participants were informed that the pictures would repeat a few times before each block ended. Participants were also informed that they could move during the breaks between the 4–min runs. Experimenters asked participants if they had any questions then turned out the lights for the task and concurrent fNIRS measurements.

#### 2.2.1. Blocked Cyclic Picture Naming Task

Blocked cyclic picture naming is an experimental paradigm that is well established to recruit executive functions during picture naming based on prior fMRI and EEG studies [13,17]. In blocked cyclic naming, four pictures are initially presented. Those four pictures are then repeated four times in four different orders. In total, each naming block consists of 16 naming trials of four pictures. Each picture was displayed for a set period of two seconds, so the total block lasted 32 s. Each 32 s naming block was followed by a 20 s baseline block displaying only a fixation cross (see Figure 1 for an illustration). There were a total of eight naming blocks, of which four presented pictures of semantically related words and four presented pictures of phonologically related words. Following previous studies using blocked cyclic naming paradigms, semantically related words were defined as those that belong to the same category (e.g., furniture, clothing, vegetables), and phonologically related words were defined as those that start with the same initial consonant (e.g., puzzle, pillow, perfume, piano). No categories, initial consonants or words were repeated from block to block.

#### 2.2.2. Blocked Cyclic Picture–Word Matching Task

Blocked cyclic picture–word matching has been used in fMRI studies, though less commonly than blocked cyclic naming, to examine the neural correlates of semantic relatedness in single word comprehension [12]. Analogous to blocked cyclic picture naming, in blocked cyclic picture–word matching, we presented an auditory word at the same time as four pictures were presented (one match and three foils), as shown in Figure 2. The participant provided a touchscreen response to select the picture that matched the auditory word presented.

### 2.3. NIRS System and Acquisition

Data were acquired using a multichannel continuous–wave CW7 NIRS system (TechEn Inc., Milford, MA, USA) using laser diodes at 690 and 830 nm and an acquisition frequency of 50 Hz. The CW7 has 32 frequency-encoded lasers (half at 690 and half at 830 nm) and 32 avalanche photo-diode detectors. Light is carried from the CW7 system to the probe on the head via optical fibers and received from the probe back to the instrument via detector fiber bundles.

### 2.4. Probe

The probe was designed using the AtlasViewer software, v2.11.3 [39]. The probe consisted of 10 sources and 20 detectors. Sixteen detectors were long-separation (30 mm) and four were short–separation (8 mm) from the source. The probe configuration resulted in 36 recording channels, of which 32 were long and 4 were short. The probe covered the frontal cortex, including 6 recording channels covering bilateral middle frontal cortex and 7 recording channels for slightly denser measurements over bilateral inferior frontal gyri. The probe is shown in Figure 3.

To guide probe placement, we first measured each participant’s nasion–inion (Nz-Iz) distance and the left–right preauricular (LPA–RPA) distance. Nz–Iz and LPA–RPA distances were halved to locate each midpoint to mark the Cz landmark. The Cz point marked on the cap was aligned to the Cz point on the participant’s head, then the cap was lowered and secured with a strap under the chin. Nz–Iz and LPA–RPA distances were then checked a second time once the cap was placed. Minor adjustments to cap placement were made as needed. Due to COVID–19 pandemic return-to-research restrictions on the amount of time allowed for human–to–human contact at the time these data were collected, optode location digitization was not feasible.

### 2.5. NIRS Data Processing

The fNIRS data were processed using the open-source Homer3 software [40]. The following processing pipeline was applied to the picture naming and picture–word-matching data, following current best practices [41]. First, channels were automatically and objectively pruned using the following parameters: dRange $1\times 10{}^{3}$ to $1\times 10{}^{7}$ SNR threshold 5, SD range 0 to 45. Next, intensity was converted to optical density (OD). Following that, we applied motion correction with spline interpolation with Savitzky–Golay filtering [42]. Then, a low pass filter with a cut-off of 0.5 Hz was applied to the OD signals to remove high-frequency noise. Next, optical density was converted to HbO and HbR concentrations using the modified Beer–Lambert law with a partial pathlength factor of 6 [43]. The blocks that were objectively determined with this preprocessing stream to be affected by motion artifacts were excluded from further analysis. The hemodynamic response function (HRF) was then estimated over the time range of 0 to 40 s using the general linear model (GLM) with the least-squares method for estimating the weights of consecutive Gaussian functions. To reduce the effect of physiological interference in the hemodynamic response estimation, short-separation channels were included as regressors [44]. For completeness and transparency in reporting, Figure A1, Figure A2, Figure A3, Figure A4, Figure A5, Figure A6, Figure A7 and Figure A8 in Appendix A show mean HbO and HbR concentrations over time with standard error for each task, condition, hemisphere and channel. Figure A9 in Appendix A shows pruned channels for each of the four runs by participant with race and ethnicity, following current best practices [45,46].

To test whether HbO concentration changes over time differed by condition, the β values resulting from the GLM for baseline (−2 to 0 s) were subtracted from the β values resulting from the GLM for task (from 0 to 40 s). This procedure was repeated for the phonologically and semantically related conditions of the word production and comprehension tasks. The β values (task over baseline) for each condition were then compared using pairwise *t*–tests by channel corrected for multiple comparisons using Bonferroni correction. Then, effect sizes were obtained using Cohen’s *d* to quantify the standard mean HbO change difference between the two conditions for each channel.

## 3. Results

The descriptive results, pairwise *t*–tests and Cohen’s *d* effect sizes are reported by channel corrected for multiple comparisons, first for word production, then for word comprehension.

### 3.1. Blocked Cyclic Picture Naming

Table 2 shows the mean, standard deviation and Cohen’s *d* effect sizes by condition for each channel. Figure 4 shows the mean HbO concentration changes for the blocked cyclic picture naming task by condition for each channel and region of interest. The phonological condition is shown in the leftmost boxes in maroon, and the semantic condition is shown in the rightmost boxes in light blue. Channels with significantly greater HbO concentration changes over baseline are surrounded with a black box. During word production, phonologically related words led to significantly greater increases in HbO concentration in three channels covering the right DLPFC (Channels 19, 20 and 21) and in one channel covering the left DLPFC (Channel 3). Naming semantically related words did not result in any channels with significantly greater HbO concentration change compared to naming phonologically related words.

### 3.2. Blocked Cyclic Picture–Word Matching

Table 3 shows the mean, standard deviation and Cohen’s *d* effect size by condition for each channel. Figure 5 shows the mean HbO concentration changes for the blocked cyclic picture–word matching task by condition for each channel and region of interest. The phonological condition is shown in the leftmost boxes in maroon, and the semantic condition is shown in the rightmost boxes in light blue. Channels with significantly greater HbO concentration changes over baseline are surrounded with a black box. Comprehending phonologically related words led to significantly greater HbO concentration changes in two channels covering the right IFG (channels 36 and 34). Comprehending semantically related words led to significantly greater HbO concentration changes in one channel covering the right DLPFC (channel 21).

## 4. Discussion

The goal of the current study was to investigate the neural correlates of semantic and phonological relatedness in word production and comprehension to evaluate the role of the DLPFC in lexical processes. Our results showed that blocked cyclic picture naming led to significant increases in HbO concentration over time for phonologically related words compared to semantically related words. This effect was found in three contiguous channels covering the right DLPFC and one channel covering the left DLPFC. In other words, for picture naming, greater bilateral DLPFC activation was associated with naming phonologically related words compared to naming semantically related words. In blocked cyclic picture–word marching, we found significant increases in HbO concentration over time in two channels covering the right IFG for phonologically related words compared to semantically related words. In contrast, we found significant increases in HbO concentration over time in one channel of the right DLPFC for semantically related words compared to phonologically related words. First, we will discuss potential explanations for the lack of activation in the left IFG in the current study, and second, we will outline the implications of our findings for the previously discussed roles of the DLPFC in lexical processes.

In previous work, naming semantically related pictures has been associated with left IFG activation. Most of this previous work has compared naming semantically related pictures to naming unrelated pictures [12,13]. In the current study, we compared naming semantically related pictures to phonologically related pictures. This difference in comparison condition between our work and previous work may be responsible for the different findings related to left IFG activation. Indeed, previous work on naming phonologically related pictures (compared to unrelated) reported no significant left IFG activation [13]. Along similar lines, the lack of IFG activation may be interpreted as support for approaches to blocked cyclic naming that implicate incremental learning rather than competitive lexical selection, as lexical selection is associated with the IFG, but general learning mechanisms are supported by executive functions associated with the PFC. Moreover, previous work has focused on the IFG region without considering the involvement of the DLPFC.

Our study showed DLPFC activation during the production of phonologically related words and the comprehension of semantically related words. These findings go against the hypothesis that inhibitory control over competitive lexical access drives the recruitment of the DLPFC, because this approach would predict DLPFC activation in lexical access across the production and comprehension of semantically related words. Given that inhibition of lexically competing words may not be the central role of the DLPFC in lexical processes, it is worth revisiting the wide range of alternative roles of the DLPFC related to executive functions such as planning and working memory [24,25,26]. The finding that DLPFC activation was associated with the production of phonologically related words but the comprehension of semantically related words suggests a nuanced approach that dissociates phonological and semantic control mechanisms to some degree, (e.g., [47,48]) in contrast to domain-general accounts [49,50,51].

Producing words, in contrast to comprehending words, requires elaborate timing functions to produce articulatory movements that result in a smooth speech signal integrated with the time requirements of cognitive processes such as lexical access or syntactic processing [28,33]. In our view, this approach, rather than others that implicate detecting novelty of incoming information [30,31] or conflict detection, resolution and adaptation [32], provides a potential explanation for the results obtained in this study. Some recent work may shed light on why producing phonologically but not semantically related words may have played a role in greater activation of the DLPFC. Executive control of phonological processes has been associated with the maintenance of inner speech cues in working memory in contexts with increased phonological costs, while semantic control may be more reflective of domain-general mechanisms [47,52]. Inner speech cueing is considered a performance supplementing strategy in which a phonological cue is held in working memory to facilitate the processing of an upcoming target word [52]. In the context of this study, it suggests that participants may have been maintaining inner-speech-based self-cues (phonological representations such as the initial sound of the word) in working memory for conditions in which they are asked to name pictures of words that begin with the same sound. In contrast, holding a semantic category cue in working memory would inhibit, rather than facilitate, the processing of an upcoming target, given the results of the previously discussed body of research showing increased naming latencies, associated with increased cognitive effort, in lexical access for semantically related words [12,15,16].

The relationship between executive function and language in the brain is multifaceted and potentially task-dependent. This study has not found evidence for the inhibitory control over the competitive lexical access approach to the role of executive function in producing and comprehending words. Our discussion of the results has pointed toward a role of the DLPFC in language related to planning, executing and coordinating articulation and with lexical access, and potentially inner speech cueing as a performance-supporting strategy. While these potential explanations begin to disentangle some of the complex relationships between executive function and lexical processes, they do not address the added executive control demands inherent in everyday conversation, in which speakers must also sequence words into syntactically well-formed utterances while planning their own utterances and simultaneously interpreting those of their conversation partner in addition to interpreting incoming nonverbal communication. Brain mechanisms that complement and support the canonical language network play a fundamental role in natural language communication, processes which may break down without these supports [3,4]. More future research is needed to elucidate the role of DLPFC in language use and understanding, which will, in turn, improve the evidence base for evaluating and treating conditions that may affect the executive function–language relationship (such as stuttering, aphasia, ADHD and autism).

## 5. Limitations

Several limitations of the current study should be noted. First, previous work using blocked cyclic paradigms has varied in number of participants. Wang et al. reported on 32 participants [17]. Harvey and Schnur examined data from 31 participants in their first experiment and 20 participants in their second experiment [12], and Schnur et al. had 12 participants [13]. Though previous studies have reported robust effects with small numbers, the current study has a small N (21), so the results should be interpreted with caution. With a sample of 21 participants and an alpha level of 0.05, the resulting power is low at 0.46. Additionally, behavioral data in the form of reaction time were not collected, so the current study is unable to link neural patterns with behavioral measures. Future work should more closely examine the connections between DLPFC activation and behavioral measures of language use and understanding.

## 6. Summary

In this study, we used fNIRS to examine frontal cortex hemodynamics by comparing changes in HbO concentrations in blocked cyclic picture naming and blocked cyclic picture–word matching with semantically and phonologically related words. The results showed significantly increased HbO concentration changes in the left and right DLPFC during blocked cyclic naming of phonologically related pictures. In the blocked cyclic picture–word-matching task, we found activation in the right DLPFC during the comprehension of semantically related words and activation in the right IFG during the comprehension of phonologically related words.

The finding that the DLPFC plays an outsized role in the production of phonologically but not semantically related words does not support the hypothesis that the DLPFC subserves inhibitory attentional control during competitive lexical access. We discussed two alternative potential roles of the DLPFC in supporting lexical processing: (1) planning and executing articulatory movements while coordinating articulation with lexical access; and (2) maintaining inner-speech-based cues in working memory to facilitate the production of phonologically related upcoming target words. These results may inform future clinical neuroimaging studies, as picture naming and picture–word-matching tests are important tools in the assessment of cognitive and communication disorders [34].

## Figures and Tables

**Figure 1 brainsci-13-01113-f001:**
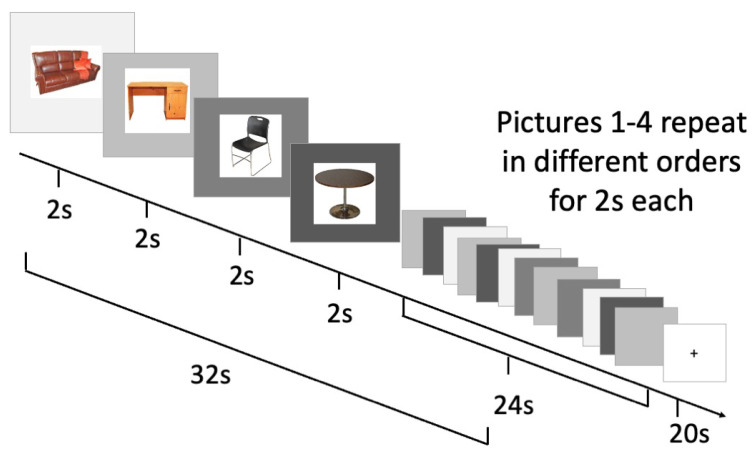
Blocked cyclic picture naming task and timing.

**Figure 2 brainsci-13-01113-f002:**
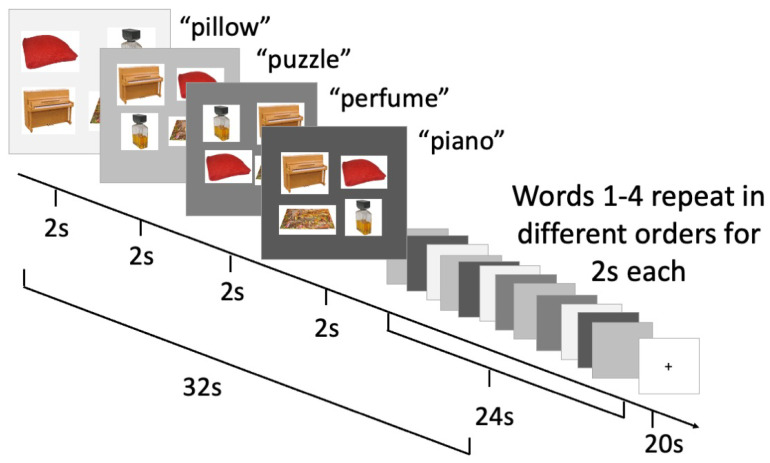
Blocked cyclic picture–auditory word matching task and timing.

**Figure 3 brainsci-13-01113-f003:**
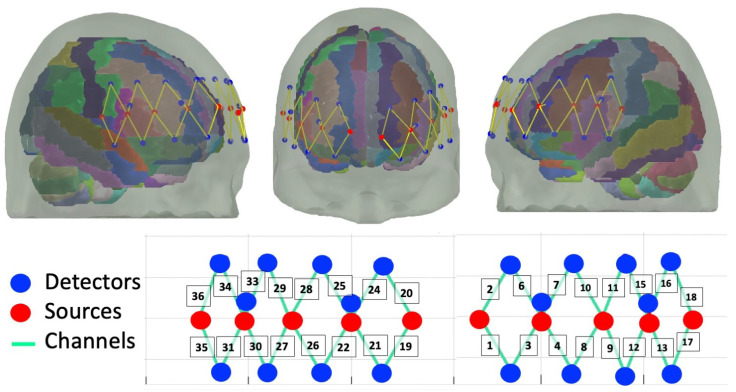
Probe design with 32 long channels and 4 short channels covering bilateral DLPFC and IFG.

**Figure 4 brainsci-13-01113-f004:**
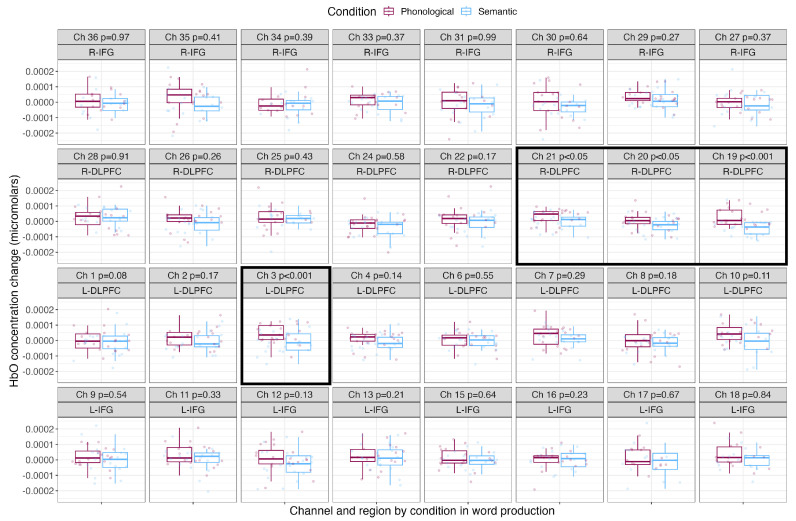
Mean HbO concentration changes over time during the production of phonologically and semantically related words. The phonological condition is shown in the leftmost boxes in maroon, and the semantic condition is shown in the rightmost boxes in light blue. Points show each participant’s individual mean HbO concentration change, with boxes showing the mean and standard deviation. P–values resulting from pairwise *t*–tests corrected for multiple comparisons are shown along with the channel number and brain region, and channels with significantly greater HbO concentration changes over baseline are outlined with a black box.

**Figure 5 brainsci-13-01113-f005:**
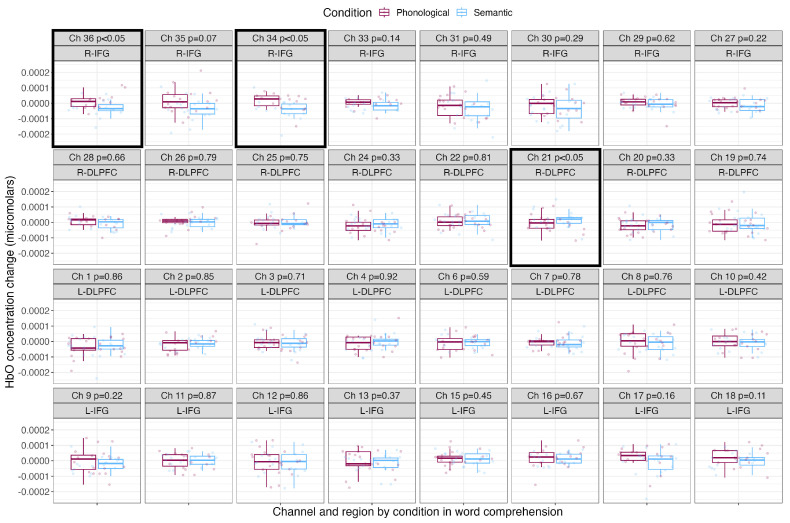
Mean HbO concentration changes over time during the comprehension of phonologically and semantically related words. The phonological condition is shown in the leftmost boxes in maroon, and the semantic condition is shown in the rightmost boxes in light blue. Points show each participant’s individual mean HbO concentration change, with boxes showing the mean and standard deviation. P–values resulting from pairwise *t*–tests corrected for multiple comparisons are shown along with the channel number and brain region, and channels with significantly greater HbO concentration changes over baseline are outlined with a black box.

**Table 1 brainsci-13-01113-t001:** Participant demographics.

Characteristic	Number (%)			
**Race and Ethnicity**	**Not Hispanic or Latino**	**Hispanic or Latino**	**Not Reported**	**Total**
Asian	4 (21%)	0	0	4 (19%)
Black or African American	1 (5.3%)	0	0	1 (4.8%)
White	12 (63.2%)	2 (100%)	0	14 (66.7%)
More than one race	2 (10.5%)	0	0	2 (9.5%)
Total	19 (90.5%)	2 (9.5%)	0	21 (100%)

**Table 2 brainsci-13-01113-t002:** Mean HbO concentration change (micromolars) by condition (phonological, semantic) for each channel in the word production task and Cohen’s *d* effect size for each channel.

Ch.	M (SD) Phonological	M (SD) Semantic	Cohen’s *d*
1	$2.92\times 10{}^{-5}\left(1.09\times 10{}^{-4}\right)$	$-1.54\times 10{}^{-5}\left(7.89\times 10{}^{-5}\right)$	0.49
2	$1.25\times 10{}^{-5}\left(6.01\times 10{}^{-5}\right)$	$-9.61\times 10{}^{-6}\left(6.34\times 10{}^{-5}\right)$	0.37
3	$1.26\times 10{}^{-5}\left(6.01\times 10{}^{-5}\right)$	$-1.12\times 10{}^{-5}\left(6.95\times 10{}^{-5}\right)$	0.88
4	$5.85\times 10{}^{-6}\left(5.39\times 10{}^{-5}\right)$	$-1.27\times 10{}^{-5}\left(5.20\times 10{}^{-5}\right)$	0.39
6	$2.25\times 10{}^{-6}\left(5.8\times 10{}^{-5}\right)$	$-4.47\times 10{}^{-6}\left(5.64\times 10{}^{-5}\right)$	0.12
7	$2.6\times 10{}^{-5}\left(6.82\times 10{}^{-5}\right)$	$9.77\times 10{}^{-6}\left(6.16\times 10{}^{-5}\right)$	0.26
8	$-1.98\times 10{}^{-5}\left(1.02\times 10{}^{-4}\right)$	$-2.13\times 10{}^{-5}\left(5.12\times 10{}^{-5}\right)$	0.02
9	$1.51\times 10{}^{-5}\left(6.18\times 10{}^{-5}\right)$	$6.89\times 10{}^{-7}\left(9.45\times 10{}^{-5}\right)$	0.19
10	$4.23\times 10{}^{-5}\left(6.30\times 10{}^{-5}\right)$	$-7.74\times 10{}^{-6}\left(8.63\times 10{}^{-5}\right)$	0.68
11	$3.08\times 10{}^{-5}\left(7.37\times 10{}^{-5}\right)$	$1.21\times 10{}^{-6}\left(7.76\times 10{}^{-5}\right)$	0.40
12	$1.21\times 10{}^{-5}\left(8.79\times 10{}^{-5}\right)$	$-3.37\times 10{}^{-5}\left(8.47\times 10{}^{-5}\right)$	0.55
13	$-3.40\times 10{}^{-5}\left(1.91\times 10{}^{-4}\right)$	$1.24\times 10{}^{-5}\left(8.49\times 10{}^{-5}\right)$	0.32
15	$-1.19\times 10{}^{-5}\left(1.31\times 10{}^{-4}\right)$	$-7.25\times 10{}^{-7}\left(5.14\times 10{}^{-5}\right)$	0.12
16	$-2.86\times 10{}^{-5}\left(1.35\times 10{}^{-4}\right)$	$1.65\times 10{}^{-5}\left(8.98\times 10{}^{-5}\right)$	0.41
17	$-2.27\times 10{}^{-5}\left(1.4\times 10{}^{-4}\right)$	$-5.42\times 10{}^{-6}\left(1.08\times 10{}^{-4}\right)$	0.14
18	$8.71\times 10{}^{-6}\left(1.28\times 10{}^{-4}\right)$	$2.95\times 10{}^{-6}\left(6.77\times 10{}^{-5}\right)$	0.06
19	$2.18\times 10{}^{-5}\left(5.24\times 10{}^{-5}\right)$	$-3.69\times 10{}^{-5}\left(5.67\times 10{}^{-5}\right)$	1.22
20	$1.42\times 10{}^{-6}\left(5.51\times 10{}^{-5}\right)$	$2.48\times 10{}^{-5}\left(4.83\times 10{}^{-5}\right)$	0.59
21	$2.67\times 10{}^{-5}\left(4.75\times 10{}^{-5}\right)$	$-3.43\times 10{}^{-6}\left(5.05\times 10{}^{-5}\right)$	0.65
22	$1.85\times 10{}^{-5}\left(8.42\times 10{}^{-5}\right)$	$-1.10\times 10{}^{-5}\left(5.88\times 10{}^{-5}\right)$	0.42
24	$-2.29\times 10{}^{-5}\left(6.93\times 10{}^{-5}\right)$	$-3.31\times 10{}^{-5}\left(5.67\times 10{}^{-5}\right)$	0.17
25	$2.63\times 10{}^{-5}\left(7.46\times 10{}^{-5}\right)$	$7.15\times 10{}^{-6}\left(6.37\times 10{}^{-5}\right)$	0.29
26	$7.81\times 10{}^{-6}\left(5.94\times 10{}^{-5}\right)$	$-5.00\times 10{}^{-6}\left(6.66\times 10{}^{-5}\right)$	0.21
27	$-2.30\times 10{}^{-5}\left(1.01\times 10{}^{-4}\right)$	$-7.54\times 10{}^{-6}\left(8.17\times 10{}^{-5}\right)$	0.18
28	$2.54\times 10{}^{-5}\left(8.22\times 10{}^{-5}\right)$	$2.94\times 10{}^{-5}\left(5.43\times 10{}^{-5}\right)$	0.06
29	$2.83\times 10{}^{-5}\left(4.77\times 10{}^{-5}\right)$	$1.72\times 10{}^{-5}\left(5.72\times 10{}^{-5}\right)$	0.22
30	$-2.09\times 10{}^{-5}\left(1.31\times 10{}^{-4}\right)$	$-3.9\times 10{}^{-5}\left(6.36\times 10{}^{-5}\right)$	0.19
31	$-1.9\times 10{}^{-5}\left(1.17\times 10{}^{-4}\right)$	$-2.41\times 10{}^{-5}\left(7.38\times 10{}^{-5}\right)$	0.05
33	$5.19\times 10{}^{-6}\left(5.56\times 10{}^{-5}\right)$	$-8.81\times 10{}^{-6}\left(5.74\times 10{}^{-5}\right)$	0.28
34	$-7.45\times 10{}^{-6}\left(7.64\times 10{}^{-5}\right)$	$-2.95\times 10{}^{-5}\left(6.3\times 10{}^{-5}\right)$	0.35
35	$1.29\times 10{}^{-5}\left(1.42\times 10{}^{-4}\right)$	$-1.91\times 10{}^{-5}\left(9.40\times 10{}^{-5}\right)$	0.27
36	$-1.08\times 10{}^{-5}\left(1.17\times 10{}^{-4}\right)$	$-1.66\times 10{}^{-5}\left(7.21\times 10{}^{-5}\right)$	0.06

**Table 3 brainsci-13-01113-t003:** Mean HbO concentration change (micromolars) by condition (phonological, semantic) for each channel in the word comprehension task and Cohen’s *d* effect size for each channel.

Ch.	M (SD) Phonological	M (SD) Semantic	Cohen’s *d*
1	$-3.37\times 10{}^{-5}\left(5.99\times 10{}^{-5}\right)$	$-3.17\times 10{}^{-5}\left(6.71\times 10{}^{-5}\right)$	0.03
2	$-1.63\times 10{}^{-5}\left(4.82\times 10{}^{-5}\right)$	$-1.40\times 10{}^{-5}\left(3.11\times 10{}^{-5}\right)$	0.06
3	$-9.87\times 10{}^{-6}\left(5.10\times 10{}^{-5}\right)$	$-7.36\times 10{}^{-6}\left(5.17\times 10{}^{-5}\right)$	0.05
4	$-6.70\times 10{}^{-6}\left(5.6\times 10{}^{-5}\right)$	$-5.85\times 10{}^{-6}\left(5.44\times 10{}^{-5}\right)$	0.02
6	$-1.17\times 10{}^{-5}\left(5.12\times 10{}^{-5}\right)$	$-4.54\times 10{}^{-6}\left(3.84\times 10{}^{-5}\right)$	0.16
7	$-5.53\times 10{}^{-6}\left(4.60\times 10{}^{-5}\right)$	$-7.82\times 10{}^{-6}\left(4.53\times 10{}^{-5}\right)$	0.05
8	$-6.59\times 10{}^{-6}\left(7.16\times 10{}^{-5}\right)$	$-1.18\times 10{}^{-5}\left(5.27\times 10{}^{-5}\right)$	0.09
9	$3.66\times 10{}^{-7}\left(7.88\times 10{}^{-5}\right)$	$-1.97\times 10{}^{-5}\left(4.55\times 10{}^{-5}\right)$	0.32
10	$-3.11\times 10{}^{-6}\left(5.51\times 10{}^{-5}\right)$	$-9.8\times 10{}^{-6}\left(4.81\times 10{}^{-5}\right)$	0.14
11	$6.76\times 10{}^{-7}\left(5.11\times 10{}^{-5}\right)$	$1.95\times 10{}^{-6}\left(3.97\times 10{}^{-5}\right)$	0.03
12	$-4.49\times 10{}^{-6}\left(8.12\times 10{}^{-5}\right)$	$-7.66\times 10{}^{-6}\left(7.39\times 10{}^{-5}\right)$	0.04
13	$-6.58\times 10{}^{-6}\left(7.56\times 10{}^{-5}\right)$	$-2.97\times 10{}^{-5}\left(7.71\times 10{}^{-5}\right)$	0.31
15	$1.60\times 10{}^{-5}\left(4.79\times 10{}^{-5}\right)$	$9.27\times 10{}^{-6}\left(4.39\times 10{}^{-5}\right)$	0.15
16	$2.04\times 10{}^{-5}\left(6.34\times 10{}^{-5}\right)$	$1.24\times 10{}^{-5}\left(3.88\times 10{}^{-5}\right)$	0.16
17	$2.35\times 10{}^{-5}\left(6.72\times 10{}^{-5}\right)$	$-9.37\times 10{}^{-6}\left(8.39\times 10{}^{-5}\right)$	0.45
18	$2.35\times 10{}^{-5}\left(5.87\times 10{}^{-5}\right)$	$1.47\times 10{}^{-6}\left(4.58\times 10{}^{-5}\right)$	0.44
19	$-8.85\times 10{}^{-6}\left(6.34\times 10{}^{-5}\right)$	$-4.36\times 10{}^{-6}\left(6.78\times 10{}^{-5}\right)$	0.07
20	$-2.2\times 10{}^{-5}\left(4.38\times 10{}^{-5}\right)$	$-1.20\times 10{}^{-5}\left(4.88\times 10{}^{-5}\right)$	0.23
21	$-5.24\times 10{}^{-6}\left(5.25\times 10{}^{-5}\right)$	$1.86\times 10{}^{-5}\left(5.33\times 10{}^{-5}\right)$	0.46
22	$6.37\times 10{}^{-6}\left(4.41\times 10{}^{-5}\right)$	$9.15\times 10{}^{-6}\left(5.30\times 10{}^{-5}\right)$	0.06
24	$-2.10\times 10{}^{-5}\left(5.04\times 10{}^{-5}\right)$	$-1.13\times 10{}^{-5}\left(4.1\times 10{}^{-5}\right)$	0.22
25	$-3.11\times 10{}^{-6}\left(5.08\times 10{}^{-5}\right)$	$1.89\times 10{}^{-6}\left(4.09\times 10{}^{-5}\right)$	0.11
26	$-6.96\times 10{}^{-6}\left(7.87\times 10{}^{-5}\right)$	$-1.21\times 10{}^{-6}\left(3.45\times 10{}^{-5}\right)$	0.10
27	$1.14\times 10{}^{-6}\left(3.59\times 10{}^{-5}\right)$	$-1.56\times 10{}^{-5}\left(4.05\times 10{}^{-5}\right)$	0.45
28	$3.34\times 10{}^{-6}\left(3.75\times 10{}^{-5}\right)$	$-1.93\times 10{}^{-6}\left(4.32\times 10{}^{-5}\right)$	0.14
29	$3.23\times 10{}^{-6}\left(4.46\times 10{}^{-5}\right)$	$-1.98\times 10{}^{-6}\left(3.02\times 10{}^{-5}\right)$	0.15
30	$-1.45\times 10{}^{-5}\left(6.61\times 10{}^{-5}\right)$	$-3.69\times 10{}^{-5}\left(8.36\times 10{}^{-5}\right)$	0.31
31	$-1.89\times 10{}^{-5}\left(6.92\times 10{}^{-5}\right)$	$-4.13\times 10{}^{-5}\left(8.87\times 10{}^{-5}\right)$	0.3
33	$1.37\times 10{}^{-6}\left(3.64\times 10{}^{-5}\right)$	$-1.28\times 10{}^{-5}\left(3.74\times 10{}^{-5}\right)$	0.4
34	$1.06\times 10{}^{-5}\left(5.69\times 10{}^{-5}\right)$	$-3.95\times 10{}^{-5}\left(6.69\times 10{}^{-5}\right)$	0.84
35	$1.38\times 10{}^{-5}\left(7.96\times 10{}^{-5}\right)$	$-3.43\times 10{}^{-5}\left(7.57\times 10{}^{-5}\right)$	0.64
36	$1.19\times 10{}^{-5}\left(4.81\times 10{}^{-5}\right)$	$-3.05\times 10{}^{-5}\left(5.57\times 10{}^{-5}\right)$	0.85

## Data Availability

The data generated in this study are publicly available through the Open Science Foundation at: osf.io/c9dbr accessed on 20 July 2023.

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
