# Peer review of "The Role of the Dorsolateral Prefrontal Cortex in the Production and Comprehension of Phonologically and Semantically Related Words"

_brainsci, 2023, doi:10.3390/brainsci13071113_

Round 1

Reviewer 1 Report

In this manuscript, Butler et al., investigated the role of dorsolateral PFC in the production and comprehension of phonologically and semantically related words using fNIRS. Authors found oversized activation of DLPFC in producing phonologically related words and suggests recruitment of executive control for maintaining inner speech cues in working memory to facilitate the production of upcoming phonologically related words. While the manuscript is well structured and written, I do have a few serious concerns.

1.    Authors hypothesized that “If DLPFC was implicated in both lexical production and comprehension, this would provide evidence for the inhibitory control competitive lexical selection approach to the role of DLPFC in language. If, however, DLPFC was not involved in both domains, alternative roles of DLPFC in lexical processing should be explored.” This is not a complete hypothesis. What are the alternative roles? This is especially important given that the results of the current study did not support the hypothesis of the inhibitory control over lexical access role of DLPFC. This means that the authors did not find anything new or scientifically significant. While the interpretation of the alternative role of DLPFC (executive control for maintaining inner speech cues in working memory to facilitate the production of upcoming phonologically related words) is reasonable, but other research has shown this role, and the current paradigm was not designed to test this role. Therefore, I question the novelty and the significance of this research.

2.    I also have concerns about the quality control of the experiment. Firstly, the results failed to replicate the classical left IFG activation in the semantically related words condition. This may be a hint that something went wrong with the data. Also, the sample size (21 subjects) is relatively too small for a neuroimaging experiment. Taken together, the results of the current study should be interpreted carefully. Whether the authors tested what they wanted to test and whether the results were robust?

3.    There are many minor errors throughout the manuscript. For example, in line 68, two back brackets. In line 73 the reference was not included in any sentence, same in line 153. From page 25 to page 28, duplicate interpretation of * and p values.

Author Response

See attached word document.

Reviewer 2 Report

Major issues:

1) naming latencies in all experiments were not analyzed. If data are available, such analysis should be performed in order to replicate data from studies that have been cited in the introduction and lead to the scientific questions addressed in this study. In addition, with these data a very valuable correlation between latency and involvement of DLPFC / IFG could be provided. If data are not available, authors should explicit explain the decision and the limitation of this study in terms of behavioral analysis.

2) Figures 4-9. The box plots are far from being readable. Construction of figures should be redone: tick labels can be shorter (Ch. instead of channel), and the plots from each region should occupy 2 rows (2x2) to better show individual points and box plots separately.

3) Figures 4-7. Beside the visualization problem discussed above, baseline SD seems 0. If this is due to subtraction of baseline to task values, representation of baseline is useless, thus just state that the task values are baseline-subtracted and show only box plots of task values.

If not, are the scattered points referred to both task and baseline values? This would result in much higher SDs... In addition, using the same color (which is the baseline color by the way) is very confounding. If all scattered points are from task values, use the red color (task color) and do not spread those values above and below the zero baseline. However, the best option would be as stated above: remove the baseline if task values are baseline-subtracted. 

Minor issues:

there are many typos for parentheses

Figure 9 legend is wrong

187-191 This paragraph seems out of place as it does not provides argumentations for using fNIRS

332 moor plans

364-365 change "this result aligns" to "our results agree with" to avoid confusion with the previous sentence that specifically referred to semantically related words.

433 rephrase "much also sequence..."

No comments

Reviewer 3 Report

Appraisal: Authors can find my  concerns, as follow:

According to me, the Introduction and its paragraphs need to be summarized. Indeed, I appreciate the effort of the authors in describing the role played by IFG and DLPFC in language processing and it was really interesting. However, it is not a review and you can find some specific suggestions:

The sentence in the introduction at line 60 – “These proposals all converge on a central theme, that DLPFC is involved in more cognitively demanding language tasks” is quite obvious for a neuroscientist, maybe less for a student, but I advise to delete it or reformulate.

“Line 87 Few studies, however, have directly compared production and comprehension tasks.” This section seems to force the role played by DLPFC in language processing since the authors did not put a rationale about DLPFC. I advise to specify the role of DLPFC found in previous studies and then introduce it.

Line 112- In the context of the current study, the lack of left IFG activation during the  naming of phonologically related pictures and the whole-brain analysis of Schnur et al.(2009) suggests that a closer examination of the role of other regions of the frontal cortex in lexical processing, specifically DLPFC, is needed- Please, reformulate this sentence.

EEG/ERP Studies: According to me, this part of the manuscript should contain results about the focus of the paper. Please reformulate this part and contextualize it.

The authors wrote a very exhaustive introduction, but the aims of the study needs to be rewritten in a more specific way.

Results: Unfortunately, I did not read in which manner authors performed statistical analysis. This Section needs to be written in a better way, with the tests used, the p values, the effect size etc.

Round 2

Reviewer 1 Report

I understand that this study was designed to examine evidence for or against the inhibitory control over lexical access hypothesis. As far as I am concerned, this study was not a typical “null results” study. Authors did find activations in DLPFC. It is just that the activation patterns were not how authors hypothesized. Therefore, I am not against publishing null results. In fact, studies with null findings that have a strong research design and use rigorous methods with appropriate statistical analyses should be part of the evidence base.

Authors found activation patterns that were “against” the inhibitory control role of DLPFC. However, that does not necessarily mean or suggest the alternative role of DLPFC being valid. New data or another experiment are needed to test the new hypothesis in order to make the point convincingly. However, the way the current abstract and discussion addressed it as if the alternative role was the major finding of the current study. This and the fact that other research has shown this role already (executive control for maintaining inner speech cues in working memory to facilitate the production of upcoming phonologically related words) make me question the novelty and the significance of the current study.

Also, a more proper way to justify the sample size issues is to conduct a power analysis for sample size. And simply adding “(and will be discussed more below)” in line 92 without indicating where confuses readers.

Author Response

I understand that this study was designed to examine evidence for or against the inhibitory control over lexical access hypothesis. As far as I am concerned, this study was not a typical “null results” study. Authors did find activations in DLPFC. It is just that the activation patterns were not how authors hypothesized. Therefore, I am not against publishing null results. In fact, studies with null findings that have a strong research design and use rigorous methods with appropriate statistical analyses should be part of the evidence base.

RESPONSE: We thank the reviewer for their time and commentary on this manuscript. The attached pdf shows our changes in blue font, and major points are addressed below.

Authors found activation patterns that were “against” the inhibitory control role of DLPFC. However, that does not necessarily mean or suggest the alternative role of DLPFC being valid. New data or another experiment are needed to test the new hypothesis in order to make the point convincingly. However, the way the current abstract and discussion addressed it as if the alternative role was the major finding of the current study. This and the fact that other research has shown this role already (executive control for maintaining inner speech cues in working memory to facilitate the production of upcoming phonologically related words) make me question the novelty and the significance of the current study.

RESPONSE: We removed the sentence from the abstract that mentioned alternative roles of DLPFC to reduce its prominence, so it does not come across as the major finding. We revised the Summary section as well to reduce the prominence of the alternative roles so they do not come across as a major finding but are rather a topic of post-hoc discussion.

Also, a more proper way to justify the sample size issues is to conduct a power analysis for sample size. And simply adding “(and will be discussed more below)” in line 92 without indicating where confuses readers.

RESPONSE: We removed “(and will be discussed more below)” so as to not be a distraction or source of confusion. We added to the Limitations section a power calculation based on this sample size.

Reviewer 2 Report

Authors addressed the raised issues.

No comment.

Author Response

Authors addressed the raised issues.

RESPONSE: We thank the reviewer for their time and commentary on this manuscript. The attached pdf shows our changes in blue font.

Reviewer 3 Report

I found the manuscript improved, however, the effect size for t test can be calculated using Cohen’s d. Effect size is important for meta-analyses.

Please follow the article from Daniel Lakens about this  : https://www.frontiersin.org/articles/10.3389/fpsyg.2013.00863/full

Author Response

I found the manuscript improved, however, the effect size for t test can be calculated using Cohen’s d. Effect size is important for meta-analyses. 

Please follow the article from Daniel Lakens about this  : https://www.frontiersin.org/articles/10.3389/fpsyg.2013.00863/full

RESPONSE: We thank the reviewer for their time and commentary on this manuscript. The attached pdf shows our changes in blue font. We have added Tables 2 and 3 to include the mean changes in oxygenated hemoglobin concentrations, standard deviations, and the Cohen’s d effect sizes.
